# Evaluation of the Efficacy of a Lift-Assist Device Regarding Caregiver Posture and Muscle Load for Transferring Tasks

**DOI:** 10.3390/ijerph20021174

**Published:** 2023-01-09

**Authors:** Yong-Ku Kong, Kyeong-Hee Choi, Sang-Soo Park, Jin-Woo Shim, Hyun-Ho Shim

**Affiliations:** 1Department of Industrial Engineering, Sungkyunkwan University, Suwon 16419, Republic of Korea; 2Digital Healthcare R&D Department, Korea Institute of Industrial Technology, Cheonan 31056, Republic of Korea

**Keywords:** caregiver, lifting task, electromyography, motion capture, work-related musculoskeletal disorders

## Abstract

The aim of this study was to confirm the effect of a lift-assist device when performing a patient-lifting task. Ten working caregivers participated in this experiment, and lifting patients from bed to wheelchair (B2C) and wheelchair to bed (C2B) was performed for manual care (MC) and lift-assist device (robot) care (RC). EMG sensors and IMU motion sensors were attached as indicators of the assistive device’s effectiveness. EMG was attached to the right side of eight muscles (UT, MD, TB, BB, ES, RF, VA, and TA), and flexion/extension angles of the neck, shoulder, back, and knee were collected using motion sensors. As a result of the analysis, both B2C and C2B showed higher muscle activities in MC than RC. When using a lift-assist device to lift patients, the RC method showed reductions in muscle activities compared to MC. As a result of the work-posture analysis, both the task type and the task phase exhibited pronounced reductions in shoulder, back, and knee ROM (range of motion) compared to those of MC. Therefore, based on the findings of this study, a lift-assist device is recommended for reducing the physical workloads of caregivers while performing patient-lifting tasks.

## 1. Introduction

A caregiver is a person that provides physical and psychological care to patients or elderly people in need, and professional caregivers are very easily exposed to work-related musculoskeletal disorders (WMSDs) [1,2]. Fifty percent of these caregivers complain of back pain each year, and 80 percent of caregivers have experienced back pain during the care period [3,4,5].

Manual patient handling, one caring task, is a major issue in healthcare sectors due to forceful exertion (handling overweight and obese patients), awkward posture (reaching, kneeling, bending, and twisting), and repetitions [6,7,8]. According to the survey studies by Sinha et al. [9] and Davis and Kotowski [1], caregivers frequently reported pain in the back (52%), followed by the neck (45%), shoulder (38%), and knee (26%) as the most frequently injured areas. In addition, among the various care activities, manual patient-handling tasks were highly associated with WMSDs. The patient-transferring task was the most likely to contribute to WMSDs at 75%, followed by a static posture (42%), and an excessive number of patients to care for (35%). From previous research findings, it can be seen that the physical loads of caregivers are caused by various manual transferring tasks on many body parts.

An assistive lifting system has been recommended to reduce caregivers’ physical pressure and workloads and to prevent WMSDs caused by patient transfer, such as lifting care [10,11]. In addition, various lift-assisted devices have been developed, and several validation studies have been conducted.

Keir and MacDonell [12] tested the muscle activities of the erector spinae, latissimus dorsi, and trapezius during ceiling lift, floor lift, and manual lift tasks in both novice and experienced groups. They demonstrated that lower muscle activities were generally found with the use of assistive lifting devices (ceiling and floor lifts), and the highest muscle activities were observed with the manual lift without any assistive devices. Hwang et al. [13] evaluated the effect of an air-assisted device on the upper-limb and back muscle activities during patient-turning tasks. They reported that the assistive device is an effective engineering control to reduce muscle loads. Low back muscle activity (ES) and upper-limb muscle activities (UT, MD, TB, and BB) were 39% and 21–44% lower with the use of an air-assisted turning device compared to manual turning without assistive devices. In the Brinkmann et al. [14] study, muscle activities of the erector spinae (ES), gluteus maximus (GM), biceps femoris (BF), rectus femoris (RF), and vastus medialis (VM) were analyzed during caregiving tasks with a manikin (80 kg) in a laboratory environment. They reported that the muscle activity of a lower-limb muscle (VM) was relatively high, whereas the activity of a back muscle (ES) tended to decrease by 24 to 42% in a squatting posture.

In addition to EMG analysis, there was also a study that evaluated the possibility of a potential protective effect of a patient lift system against L5/S1 (between the fifth lumbar vertebrae and top of the sacrum) compression force. Marras et al. [15] reported that operating a ceiling-mounted patient lift system would be considered safe on the lumbar spine, whereas L5/S1 compression force easily exceeded 3400N when performing manual transfer tasks.

While most of the previous studies mainly focused on the effect of reducing the workloads on the upper extremities and backs of the caregivers, there are very few studies that have evaluated workloads across the whole body or lower extremities. Therefore, the objective of this study was to analyze the body posture and the upper-limb, lower-limb, and back muscle activities when transferring patients from bed to wheelchair and wheelchair to bed manually or with the use of an assistive lift device.

## 2. Materials and Methods

### 2.1. Experimental Task

In this study, the lifting care task was first analyzed to understand the effect of reducing the workloads of caregivers with the use of a lift-assist device. The task phase was classified into a preparation phase (Pre-phase), a main lifting-task phase (Front-phase), and a finishing phase (Post-phase). A task method was defined as Manual Care (MC), where a task is performed without a lift-assist device, and Robot Care (RC), which refers to the use of a lift-assist device. B2C and tasks were evaluated.

In the B2C task, MC consists of a total of nine sub-tasks (2 Pre-phases, 5 Front-phases, and 2 Post-phases), and RC consists of a total of 17 sub-tasks (3 Pre-phases, 11 Front-phases, and 3 Post-phases). In both MC and RC task methods, caregivers performed ‘Pr1—wheelchair setting’ and ‘Pr2—handrail lowering’ for the Pre-phase, and ‘Po1—patient foot adjustment’ and ‘Po2—handrail raising’ for the Post-phase. In RC, additional sub-tasks, ‘Pr3—lift moving 1′ and ‘Po3—lift moving 3′, were performed to prepare and move a lift-assist device.

There are considerable differences between sub-tasks in MC and RC in the case of the Front-phase. In the case of MC, there are five sub-tasks comprising the Front-phase: ‘Fr1—adjusting patient posture’, ‘Fr2—wearing slippers’, ‘Fr3—sitting patient (bed), ‘Fr4—sitting patient (wheelchair)’, and ‘Fr5—deep sitting patient (wheelchair)’. On the other hand, in the case of RC, there are 11 sub-tasks in the Front-phase: ‘Fr1—sling-sheet setting’, ‘Fr2—moving and adjusting lift’, ‘Fr3—sling-sheet/lift install’, ‘Fr4—lifting patient’, ‘Fr5—moving patient (wheelchair)’, ‘Fr6—sitting patient’, ‘Fr7—sling-sheet uninstall’, ‘Fr8—lift moving 2′, ‘Fr9—sling-sheet uninstall’, ‘Fr10—deep sitting patient (wheelchair)’, and ‘Fr11—wearing slippers’ (Figure 1).

In the C2B task, MC consists of nine sub-tasks (3 Pre-phases, 5 Front-phases, and 1 Post-phase), and RC consists of a total of 15 sub-tasks (3 Pre-phases, 10 Front-phases, and 2 Post-phases). In both MC and RC task methods, caregivers perform ‘Pr1—wheelchair setting’ and ‘Pr2—handrail lowering’ for the Pre-phase. Then, ‘Pr3—patient foot adjustment’ and ‘Pr3—lift moving 1′ were performed for MC and RC. For the Post-phase, both MC and RC have ‘Po1—handrail raising’, and ‘Po2—Lift moving 3′ was performed for RC only.

Similar to the B2C, there are differences between sub-tasks in MC and RC in the case of the Front-phase. In the case of MC, there are five sub-tasks in the Front-phase: ‘Fr1—pulling patient forward’, ‘Fr2—sitting patient (bed)’, ‘Fr3—laying patient (bed)’, ‘Fr4—adjusting patient posture’, and ‘Fr5—taking off slippers’. On the other hand, in the case of RC, there are ten sub-tasks in the Front-phase: ‘Fr1—pulling patient forward’, ‘Fr2—sling-sheet setting’, ‘Fr3—moving and adjusting lift’, ‘Fr4—sling-sheet/lift install’, ‘Fr5—lifting patient’, ‘Fr6—moving patient (bed)’, ‘Fr7—laying patient (bed)’, ‘Fr8—sling-sheet uninstall’, ‘Fr9—lift moving 2′, and ‘Fr10—sling-sheet uninstall’ (Figure 2).

### 2.2. Apparatus

In this study, surface electromyography (EMG) equipment (TeleMyo 2400 DTS System) from Noraxon (Scottsdale, AZ, USA) was used to measure muscle loads when using a lift-assist device.

Four upper-limb muscles (UT, MD, TB, and BB), one back muscle (ES), and three lower-limb muscles (RF, VM, and TA) were tested in this study (Figure 3). The electrode attachment positions were distinguished by referring to SENIAM guidelines [16]. Band-pass filtering was implemented for noise reduction, with pass-band frequencies 20–400 Hz, and the sampling rate was 1500 Hz. EMGs of resting and maximum voluntary contraction (MVC) for each muscle group were measured twice for normalization [17].

The MVN Awinda system (Xsens Technology B.V., Enschede, The Netherlands), a sensor based on Inertia Measure Units (IMUs), was applied to measure body posture angles according to the task types and methods. A total of 17 sensors were attached to the head (1), sternum (1), pelvis (1), both sides of the shoulder (2), upper arm (2), lower arm (2), hand (2), upper leg (2), lower leg (2), and foot (2) to collect the flexion/extension angles of the neck, shoulder, and back, and the flexion angle of the knee, as shown in Figure 4.

### 2.3. Experimental Procedure

Ten caregivers working at nursing sites participated in this study. The average age of the participants was 54.4 (±12.8) years old, and the average height was 163.2 (±8.1) cm. All participants were incumbent caregivers and had more than eight years of experience in hospitals or facilities. A total of 10 caregivers (9 women and one man) participated in this experiment. All of them had no experience with musculoskeletal diseases in the past six months and physical defects in conducting patient-lifting tasks. This study has been approved by the Korea Institute of Industrial Technology (KITECH, IRB File No. 2022-001-002). All participants were informed about the content and purpose of this experiment and provided a consent form in advance. In addition, the participants fully understood how to use the assistive device so that they did not have difficulty caring for patients.

After measuring the basic anthropometric measurements (height, shoulder height/width, elbow span, wrist span, arm span, hip height/width, knee height, and ankle height) for motion-capture analysis, EMG and IMU sensors were attached to the subjects to collect muscle activity and body angle data. After setting all the equipment, resting EMG and MVC were measured for normalization of EMG signals, and calibration was performed for the motion-capture system.

Each subject performed a total of four tasks in random order according to the combinations of two levels of task types (B2C and C2B) and two levels of task methods (MC and RC) (Figure 5). All tasks were conducted with a dummy to standardize the patient’s weight and physical conditions. Using the dummy helped reduce patient variability by maintaining the constant weight of the lift for each of the caregivers. The dummy was 165 cm tall and weighed 57 kg. The patient’s condition was assumed to be unable to cooperate with lifting tasks.

### 2.4. Statistical Analysis

Task method (MC/RC) and task phase (Pre/Front/Post) were set as independent variables, and the muscle activities and body angles collected through the motion-capture system were set as dependent variables in order to analyze the effect of a lift-assist device on the workloads of caregivers. SPSS 25 (IBM, Armonk, NY, USA) was used for all statistical analyses, and multi-way ANOVA was performed to verify the difference between dependent variables according to each independent variable (*p* = 0.05). In addition, Tukey’s test was also performed for multiple comparisons.

## 3. Results

### 3.1. Muscle Activity Analysis for RC vs. MC–B2C

As a result of analyzing the muscle activity of caregivers during transfer assistive care, in the case of B2C (bed to wheelchair), the main effect of the task type showed that MC (manual care) muscle activities were statistically significantly larger than those of RC (robot-aided care) at the TA (10%ile) and VM (90%ile). Regarding the TA muscle, the muscle activity (10%ile) of RC was 2.0% MVC, which was about 16.7% compared to the 2.4% MVC of MC. VM muscle activity (28.8% MVC-90%ile) for RC was about 28% lower than that of MC (40.0% MVC).

Although it was not statistically significant, the average muscle activities of the back, upper limb, and lower limb muscles (MD, TB, ES, and VM) tended to be reduced by 16.5% and 22.3% (i.e., MD: 11.7% & 23.6%, TB: 10.8% & 18.2%, ES: 16.2% & 19.3%, and VM: 27.4% & 28.0%) for the 50th %ile and the 90th %ile, respectively. In the case of ES and MD for RC use, the muscle loads were reduced by 16.2~19.3% and 11.7~23.6%, respectively, compared to MC use. The reductions of the VM muscle, which mainly acts in knee-joint movement, were 27.4~28.0%, prominent compared to other muscle groups (Figure 6).

Analyzing muscle activity according to the task phase (Pre-phase, Front-phase, or Post-phase) during B2C shows that the muscle activity during the Front-phase, which consists of major transfer sub-tasks, was significantly higher than that of the Pre- and Post-phases (Figure 7). In detail, UT (50 and 90%iles), MD, BB, ES (10, 50, and 90%iles), TB (10 and 50%iles), and VM (10 and 90%iles) showed statistically higher muscle activities during the Front-phase compared to other phases.

The muscles of the upper and lower limbs (UT, MD, RF, and VM) and back (ES) in the Post-phase showed similar or larger activities than those of the Pre-phase, which were more pronounced at the 50 and 90%iles, compared to the 10%ile.

In the interaction of the task type and the task phase, the muscle activities of VM (50th and 90th %iles) and TA (50%ile) for MC were statistically significantly greater than those of RC. In other words, reducing muscle loads due to the use of a lifting assistance device seems to be the most effective in VM and TA muscles in the Front-phase. As shown in Figure 8, VM muscle activities (50th and 90th %iles) were decreased by 38.0% and 39.5%, respectively, compared to MC (17.1 and 48.4% MVCs) when performing RC (10.6 and 29.3% MVCs) in the Front-phase. In addition, TA muscle activity (50th %ile) during RC also showed a reduction of muscle loads by 29.6% compared to MC.

In addition, there was no significant change in muscle activity according to the task phase in the case of RC. However, there was significantly higher muscle activity during the Front-phase than in the other task phases in the case of MC (Figure 8).

### 3.2. Muscle Activity Analysis for RC vs. MC–C2B

In the case of C2B, analyzing muscle activity according to the task type showed statistically significant results in UT (50th %ile) and VM (50th and 90th %iles) and TB, BB, and TA (10th %ile) (Figure 9). Similar to B2C, in C2B, the muscle activities of VM (10.1 and 26.1% MVCs) during RC showed muscle loads about 34.3–35.3% lower than those of MC (15.6 and 39.7% MVCs) in the higher muscle activity ranges (i.e., 50th and 90th %iles).

Generally, the average 50th and 90th %iles of muscle activities of most muscles in the RC application tended to decrease by 13.9–16.2% compared to the average muscle activity during MC (TB: 2.0% and 7.5%, BB: 12.9% and 22.1%, ES: 14.8% and 18.3%, RF: 15.9% and 12.7%, VM: 35.3% and 34.3%, and TA: 16.1% and 1.4% for the 50th %ile and the 90th %ile, respectively). In the case of BB, ES, and RF during RC, the muscle loads compared to MC use were reduced by 12.9~22.1%, 14.8~18.3%, and 15.9~12.7%, respectively, showing a relatively more significant reduction effect than other muscles.

In addition, as a result of analyzing each muscle group according to the task phase (Pre-phase, Front-phase, and Post-phase), similar to the results of B2C, the muscle activity during the Front-phase was significantly higher than that of the other phases, as shown in Figure 10. By muscle, the UT (50th and 90th %iles); MD, BB, and VM (10th, 50th, and 90th %iles); and TB and ES (10th and 50th %iles) showed significantly higher muscle activities during the Front-phase than the Pre- and Post-phases.

In the case of UT, MD, and VM, unlike the results of B2C, muscle activity during the Pre-phase tended to be similar to or greater than that of the Post-phase. This is likely because the sub-tasks of the B2C’s Post-phase and the C2B’s Pre-phase sub-tasks are similar.

The interaction analysis of the task type and task phase showed statistically significant muscle activity in the VM (50, 90%ile) and TA (50%ile) during the Front-phase. The effect of reducing muscle load throughout C2B with a lifting device seems to be the largest for the VM and TA muscles during the Front-phase (Figure 11). The muscle loads of the VM were decreased by 44.0% (19.1% MVC vs. 10.7% MVC) for the 50th %ile and 41.4% (47.3% MVC vs. 27.7% MVC) for the 90th %ile, and the muscle loads on the TA were also decreased by 29.6% (9.8% MVC vs. 6.9% MVC) for the 50th %ile.

Similar to B2C, RC showed no significant change in muscle activity as the task progressed, but MC showed significantly higher muscle activity in the Front-phase than in other phases.

### 3.3. Body Posture Analysis for RC vs. MC–B2C

As a result of the % range of motion (ROM) analysis of body posture during B2C, the main effect of the task type was statistically significant in the back (10th and 50th %iles), neck (10th and 50th %iles), shoulder (90th %ile), and knee (10th, 50th, and 90th %iles), as shown in Figure 12.

When RC was applied, the %ROMs of the back, shoulder, and knee were significantly reduced by 20.9–32.5%, 10.1%, and 19.1–37.4%, respectively, compared to MC, whereas in the case of the neck, RC showed a significantly greater %ROM than MC, with RC showing 38.0% (10th %ile) and 18.0% (50th %ile) larger ROMs than MC, respectively.

The analysis of back posture (50th and 90th %iles) according to the task phase also showed larger ROMs in the Pre- and Post-phases than in the Front-phase. The ROMs of the Pre- and Post-phase were 34.8~49.2% and 36.8~51.2%, respectively, whereas the ROMs of Front-phase were 26.9~40.4%, as shown in Figure 13. The neck posture also showed a similar trend to the back at 10th and 50th %ile ROMs, and the %ROMs during the Post-phase (14.7~35.4%) and Pre-phase (12.8~31.3%) were higher than those of the Front-phase (9.0~27.8%).

In the knee posture analysis, the ROMs, 8.9, 26.9, and 39.2% for the 10th, 50th, and 90th %ile during the Post-phase, were highest among the other phases, followed by those of the Front-phase (5.1, 13.7, and 30.8% ROMs) and Pre-phase (4.5, 12.3, and 26.4% ROMs).

In general, the %ROM when MC was applied across all task stages was greater than when RC was used, as shown in Figure 14. In particular, in the case of the knee, there were no significant differences in % ROMs between phases in the case of RC (4.9, 4.9, and 6.1% ROMs; 12.7, 5.5, and 16.3% ROMs; 27.8, 29.4, and 29.7% ROMs for the 10th, 50th, and 90th %iles, respectively), whereas in the case of MC, the %ROMs of the knee were rapidly increased from Pre-phase to Front-phase (about 1.4 times) and from Front-phase to Post-phase (about 2.2 times).

### 3.4. Body Posture Analysis for RC vs. MC–C2B

In the case of C2B, the main effect on body posture according to the task type was statistically significant in the back (50th and 90th %iles), neck (10th, 50th, and 90th %iles), shoulder (10th %ile), and knee (10th, 50th, and 90th %iles), as shown in Figure 15.

The ROM reductions were about 30.6~36.8% and 23.3~41.4% for the back and knee, respectively, when RC was applied. On the other hand, in the case of the neck, RC showed a greater %ROM (6.2~52.8%) than MC.

In addition, the analysis of back posture (50th and 90th %iles) according to task phase showed higher %ROMs in the Pre-phase and Post-phase than in the Front-phase. On the other hand, in the posture analyses of the shoulder and knee, the results of %ROM were relatively higher in the Post-phase than in the Pre- and Front-phases (Figure 16).

The interaction between task type and task phase was significant in the neck and knees (50th %ile). The %ROM of the neck during the Pre- and Post-phases when using RC was higher than the %ROM when using MC, whereas the %ROM of the knees in all task phases when using MC was higher than the % ROM when using RC (Figure 17).

In the case of the neck, it was found that the %ROM during the Front-phase was 14.6 and 37.1% larger than the Pre- and Post-phases, respectively, when performing the MC task. However, the %ROM during the Front-phase was 12.7 and 24.6% less than the Pre- and Post-phases, respectively, when performing RC tasks.

In the case of the knees, there was no significant difference in ROM across task phases when performing RC work (11.8~13.3% ROM), but in the case of MC, it was found that the %ROM decreased by about 0.622 times and 0.617 times as the work was performed from Pre-phase to Front-phase and Post-phase, respectively.

## 4. Discussion

Muscle activity according to task type (MC vs. RC) when performing patient transfer tasks showed a tendency to decrease in both B2C (bed to wheelchair) and C2B (wheelchair to bed) cases. This means that, unlike MCs, where caregivers manually lift and transfer patients, RC lifting and patient-transfer tasks reduce the physical workload on caregivers.

Overall, compared to MC, the RC method of lifting patients using a lift-assist device showed 10.8~23.6%, 16.2%, and 27.4~28.0% reductions in upper extremity muscle (MD and TB), back (ES), and lower extremity muscle (VM) activities, respectively, for B2C. Activities of upper extremity muscles (TB and BB), back (ES), and lower extremity muscles (RF, VM, and TA) were reduced by 2.0~22.1%, 14.8~18.3%, and 1.4~35.3% during C2B, similar to the B2C results. In particular, the VM, which mainly serves as a walking and knee-joint extension in RC, showed a statistically significant reduction of muscle workload (34.3~35.3%) compared to MC in both B2C and C2B (Figure 6 and Figure 9). In addition, UT, MD, TB, BB, and TA activities also showed statistically significant reductions in C2B when lifting patients was performed using a lift-assist device.

The reductions of muscle activity in the upper limb and back muscles (ES) were similar to the previous studies [12,13,18,19,20,21]. Hwang et al. [13] evaluated upper-body muscle activities when turning a patient in a bed using a lift-assist device. They revealed that the upper-limb muscle activities (UT, MD, TB, and BB) decreased by about 21–44%, and ES activity decreased by about 39% with the use of RC. In a comparative study of three types of methods, ceiling lift, floor lift, and manual lift tasks, Keir and MacDonell [12] also reported that the muscle activity of the erector spinae, latissimus dorsi, and trapezius occurred at the lowest level when using lifting assistive devices (ceiling lift and floor lift), whereas those muscle activities were very high with a manual lifting task. Silvia et al. [18] evaluated the back compressive forces (Michigan 3D Static Strength Model and EMG of the low back) by comparing the lifting assist devices and the traditional manual technique. They reported that the compressive force decreased by about 29.8% and 84.8% at B2C and C2B, respectively, using lifting devices. Garg et al. [19,20] also tested manual tasks and three hoist-assisted devices, and Nelson et al. [21] compared manual lifting tasks and ceiling-mounted lift tasks to evaluate the reduction of back muscle stress and perceived caregivers’ comfort in a laboratory study. They reported that the lower back muscles showed less physical and perceived stress when using lift-assist devices.

In particular, the reductions of ES and VM muscle activities can be explained by the %ROM results of the back and knee of the caregivers (Figure 12 and Figure 15). Brinkmann et al. [14] mentioned that the VM was very active, and the thigh, hip, and back (ES) muscles were strengthened to compensate for lumbosacral loads in the squatting posture. In manual care (MC), the muscles of the back and legs are considered to be more active due to greater back bending, knee flexion, and shoulder flexion compared to RC using a lift-assist device in a patient-lifting posture.

On the other hand, the ROM of the neck tended to increase when performing RC in both B2C and C2B. The increased neck flexion angle was likely because the neck flexion angles in manual care (MC) were relatively small due to bending posture in the back and neck when lifting the patient, whereas RC is performed by standing in place and manually looking at the lift-assist device.

The analysis of the task type and task phase interaction clearly showed the effect of reducing the muscle activities of the VM and TA by 38.0~44.0 and 29.6% in the Front-phase when using the lift-assist device (Figure 8 and Figure 11). When using the lift-assist device, most muscle activities tended to decrease in the Front-phase, which is a major patient transfer step, in both B2C and C2B tasks. This means that the physical workload of caregivers is further reduced when using transfer care aids than when caregivers directly lift patients. The reduction of VM and TA muscle loads is significant because the physical workload on the caregivers when lifting the patient has a more significant effect on the lower body. In particular, in the case of MC, unlike the lifting tasks in the industrial fields in transferring the patient from wheelchair to bed and bed to wheelchair, vertical lifting and horizontal lifting tasks are performed simultaneously. Therefore, at this time, it is necessary to bend the back and knees more for the patient’s safety, whereas RC mainly performs lifting of the patient while in a standing posture, which is thought to show a more significant difference in the muscle activity of the lower extremities—that is, RC involves operating a lift-assist device to lift a patient while standing, so the workload reduction of lower body muscles seems remarkable. Therefore, this study considers RC to have the most significant workload relief effect in the lower extremity muscles when transporting patients.

Although it was not statistically significant, when the lift-assist device was used, the ROMs of the back and shoulder were reduced by 10.1–32.5% in B2C and 30.6–36.8% in C2B. Overall, compared to manual care (manually lifting a patient without a device), the lift-assist device reduces the degree of bending of the back, shoulders, and knees, and it can be seen that the burden of muscles is reduced accordingly.

In the case of B2C, the muscle activities of the back and lower body in the Post-phase (consisting of ‘Po1—patient foot adjustment’, ‘Po2—handrail raising’, and ‘Po3—Lift moving 3′) were much higher than those in the Pre-phase (consisting of ‘Pr1—wheelchair setting’ and ‘Pr2—handrail lowering’). These results are thought to be due to the influence of the ‘Po1—patient foot adjustment’ task in B2C’s Post-phase. In the ‘Po1′ task, the caregivers take a squatting posture to arrange the feet of the patient sitting in the wheelchair and to handle the foot support of the wheelchair and thus appear to have high knee flexion angles. As a result, the ROM (39.2% ROM) of the knee tended to be statistically significantly greater in the Post-phase than that of the Pre-phase (26.4% ROM) and Front-phase (30.8% ROM) (Figure 13).

In contrast, in the case of C2B, muscle activities in the Pre-phase, such as ‘Pr1—wheelchair setting’, ‘Pr2—handrail lowering’, ‘Pr3—patient foot adjustment’ (MC only), and ‘Pr3—lift moving 1′ (RC only), were the same or larger than those in the Post-phase [consisting of ‘Po1—handrail raising’ and ‘Po2—lift moving 3′ (RC only)]. Figure 16 also showed that the back, shoulder, and knee %ROM in the Pre-phase tended to be statistically larger than those of the Post-phase, which could also be due to the ‘Po1—handrail raising’ task being performed in the Pre-phase in the case of C2B.

Only knee posture was statistically significant in the body posture study for task type and task phase. Results showed that the knee angle changed the most in the Post-phase and Pre-phase for B2C and C2B, respectively, in the MC task.

In the Post-phase of B2C, MC and RC tasks commonly perform ‘Po1—patient foot adjustment’ and ‘Po2—handrail raising’, but an additional ‘Po3—lift moving 3′ was performed for RC. In addition, in the Pre-phase of C2B, MC and RC tasks commonly involved ‘Pr1—wheelchair setting’ and ‘Pr2—handrail lowering’, but ‘Pr3—patient foot adjustment’ and ‘Pr3—lift moving 1′ were performed in the case of MC and RC, respectively. At this time, the ‘Pr3—patient foot arrangement’ task requires a large knee angle, while ‘lift moving’ is performed standing right away, so relatively little knee flexion occurs.

There are some limitations in this current study. First, a greater number a participant may be needed to generalize the results of this study, although statistically significant results were reported in the study. Thus, it is necessary to increase the number of participants in further studies. Second, as with many other existing studies, this study mainly focuses on the reduction of workload for caregivers—that is, this study was mainly focused on the caregivers’ workload, not on the patient’s safety, comfort, or satisfaction. Therefore, it would be necessary to conduct a study focusing on patients’ perspectives regarding safety and mental comfort.

## 5. Conclusions

This study aims to confirm the reduction of physical workloads from the use of a lift-assist device when a caregiver performs patient lifting. In the case of both B2C and C2B, overall muscle activities tended to decrease when using the lift-assist device. In particular, VM and TA muscle group activities tended to significantly decrease when sub-tasks of the Front-phase were performed by the lift-assist device (RC method). It was noted that the physical loads on the lower-limb muscles were reduced by using the device.

In the working posture analysis, in both the task method (B2C and C2B) and task phase (Pre-, Front-, and Post-phase), a more pronounced ROM reduction effect on the back, shoulder, and knee was observed when a lift-assist device was used compared to manual care. The physical loads when using the lift-assist device were relatively less than those of MC.

The results of this study showed that the use of assistive devices had a positive effect on the reduction of body burden when performing a patient-transferring task. Therefore, assistive devices may be recommended for reducing caregivers’ physical burden when performing a movement of a patient manually.

## Figures and Tables

**Figure 1 ijerph-20-01174-f001:**
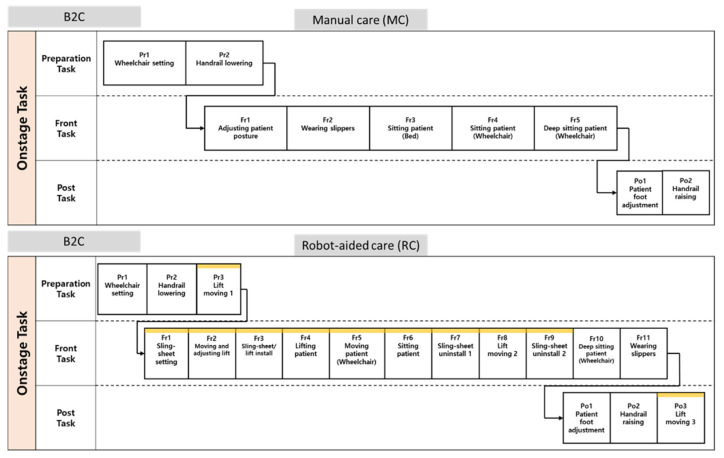
Structures of B2C sub-tasks for both MC and RC.

**Figure 2 ijerph-20-01174-f002:**
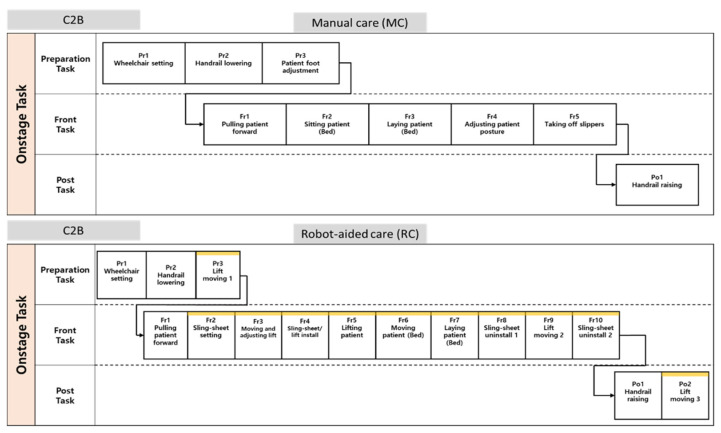
Structures of C2B sub-tasks for both MC and RC.

**Figure 3 ijerph-20-01174-f003:**
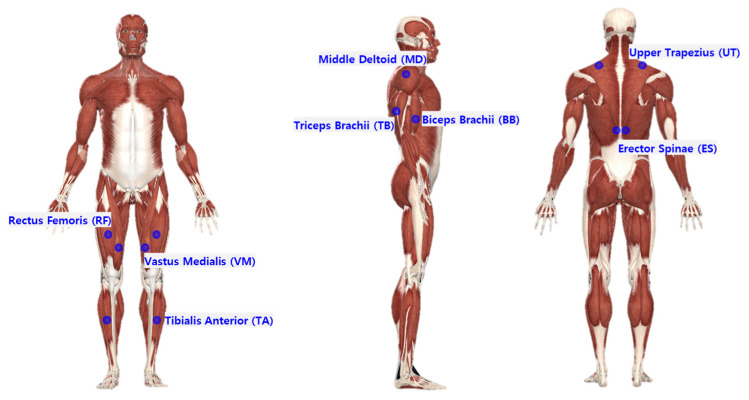
Locations of EMG sensor attachments.

**Figure 4 ijerph-20-01174-f004:**
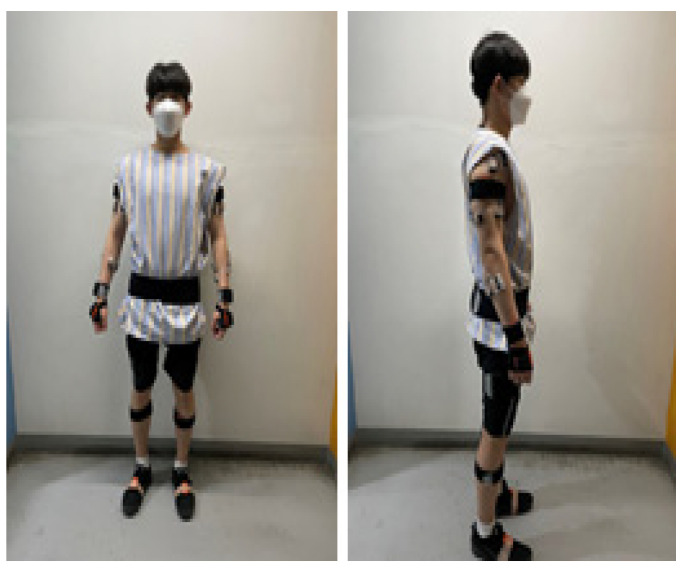
Locations of motion-capture sensor attachments.

**Figure 5 ijerph-20-01174-f005:**
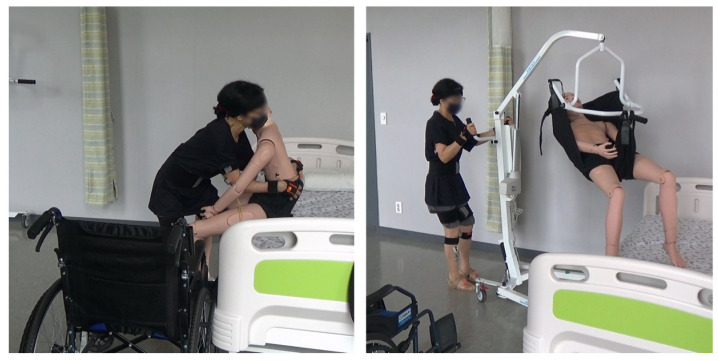
Lifting care task (**Left**: MC, **Right**: RC).

**Figure 6 ijerph-20-01174-f006:**
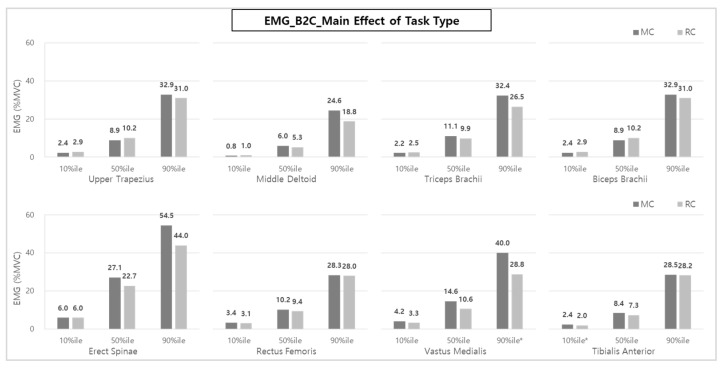
The effect of task type on each muscle activity during B2C. Asterisk (*) indicates that the difference between means is statistically significant at *p* < 0.05.

**Figure 7 ijerph-20-01174-f007:**
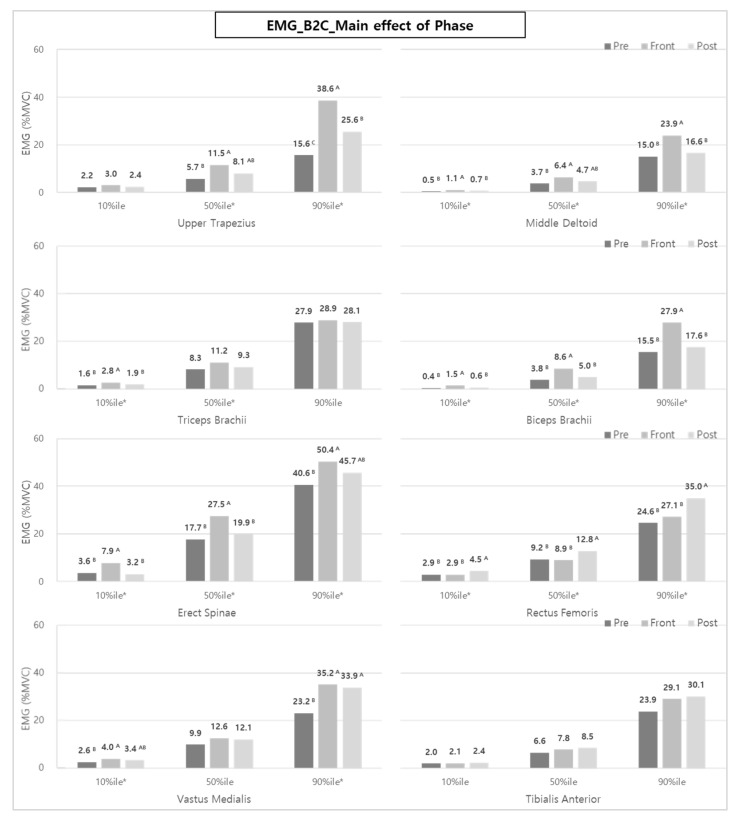
The effect of phases on each muscle activity during B2C. Asterisk (*) and the alphabet capital letter (A and B) indicate that the difference between means is statistically significant at *p* < 0.05.

**Figure 8 ijerph-20-01174-f008:**
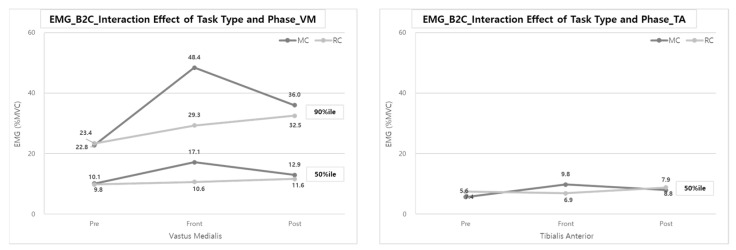
Interaction effect of task type and phase on VM & TA muscle activities during B2C.

**Figure 9 ijerph-20-01174-f009:**
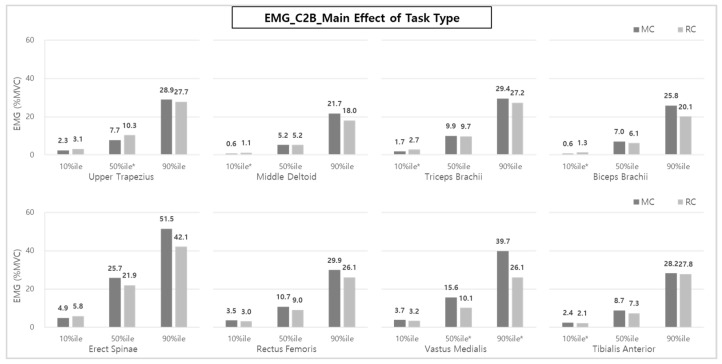
The effect of task type on each muscle activity during C2B. Asterisk (*) indicates that the difference between means is statistically significant at *p* < 0.05.

**Figure 10 ijerph-20-01174-f010:**
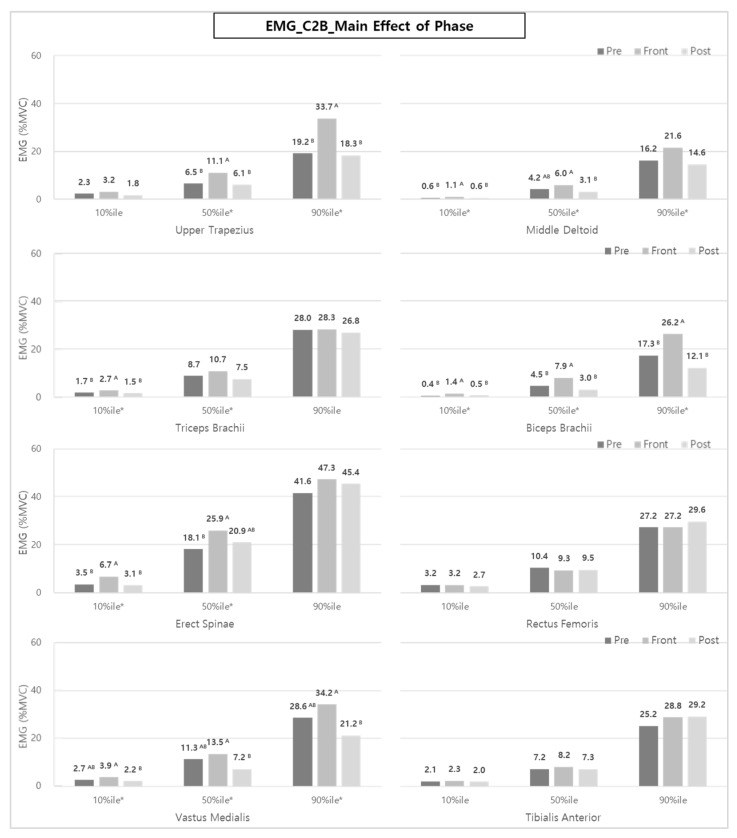
The effect of phases on each muscle activity during C2B. Asterisk (*) and the alphabet capital letter (A and B) indicate that the difference between means is statistically significant at *p* < 0.05.

**Figure 11 ijerph-20-01174-f011:**
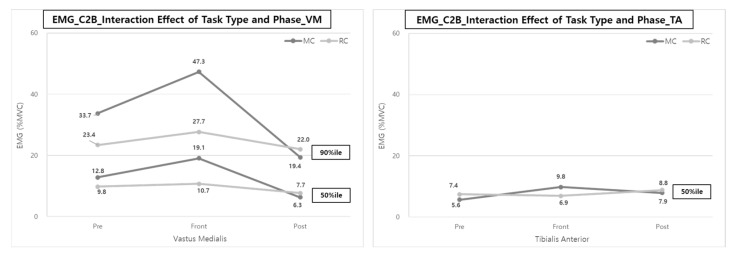
Interaction effect of task type and phase on VM & TA muscle activities during C2B.

**Figure 12 ijerph-20-01174-f012:**
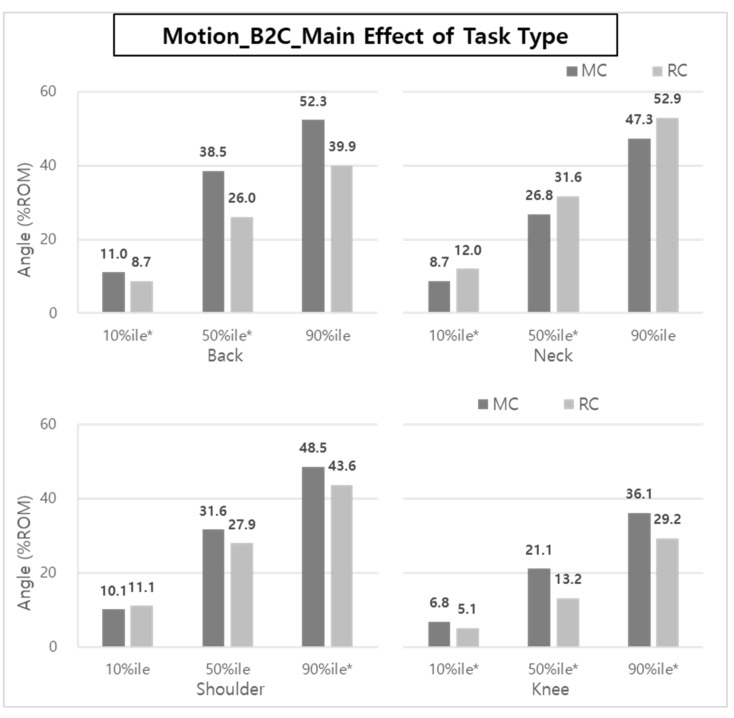
The effect of task type on body posture during B2C. Asterisk (*) indicates that the difference between means is statistically significant at *p* < 0.05.

**Figure 13 ijerph-20-01174-f013:**
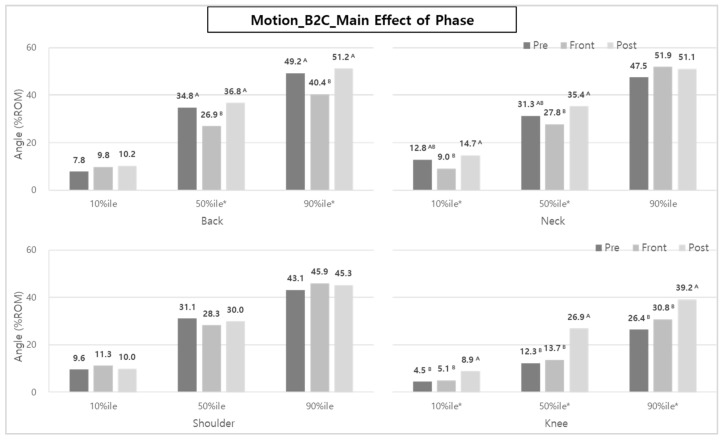
The effect of phase on body posture during B2C. Asterisk (*) and the alphabet capital letter (A and B) indicate that the difference between means is statistically significant at *p* < 0.05.

**Figure 14 ijerph-20-01174-f014:**
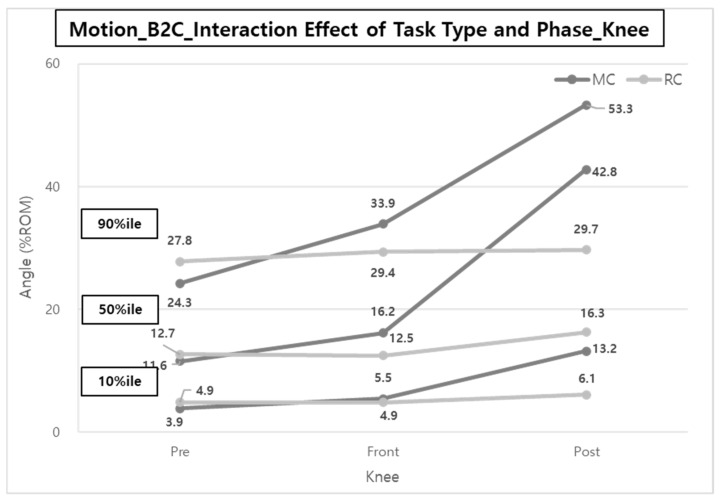
Interaction effect of task type and phase on the ROM of Knee (B2C).

**Figure 15 ijerph-20-01174-f015:**
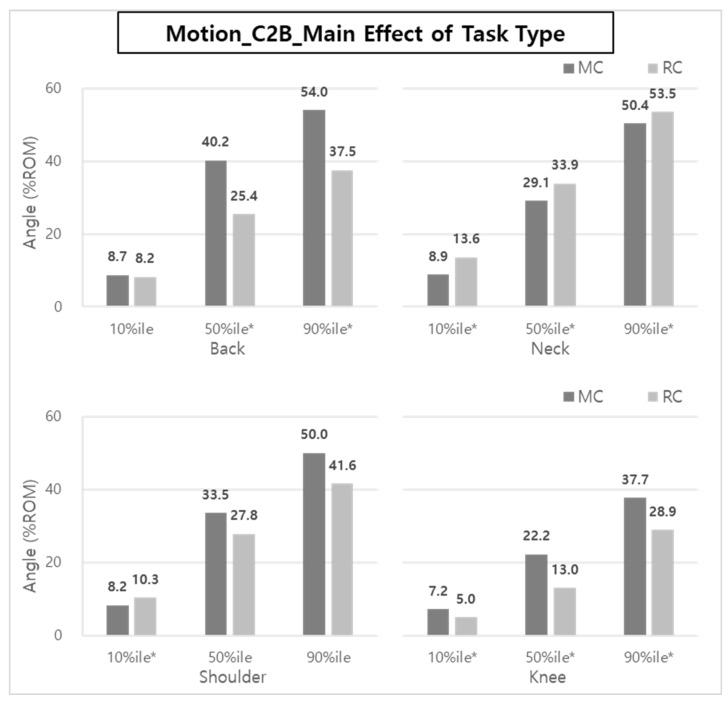
The effect of task type on body posture during C2B. Asterisk (*) indicates that the difference between means is statistically significant at *p* < 0.05.

**Figure 16 ijerph-20-01174-f016:**
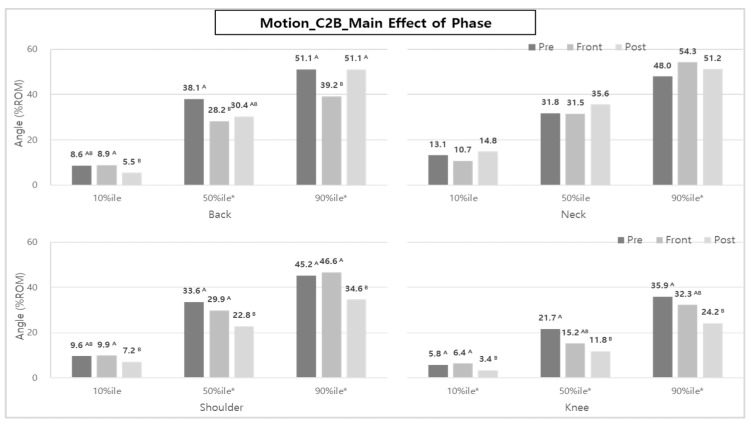
The effect of phase on body posture during C2B. Asterisk (*) and the alphabet capital letter (A and B) indicate that the difference between means is statistically significant at *p* < 0.05.

**Figure 17 ijerph-20-01174-f017:**
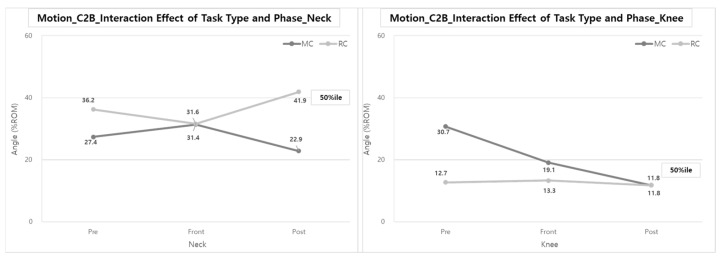
Interaction effect of task type and phase on the ROM of the Neck & Knee (C2B).

## Data Availability

Not applicable.

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
