# Peer review of "Evaluation of the Efficacy of a Lift-Assist Device Regarding Caregiver Posture and Muscle Load for Transferring Tasks"

_ijerph, 2023, doi:10.3390/ijerph20021174_

Round 1

Reviewer 1 Report

In this article, the authors emphasise the important role and help of a lifting aid for a patient. It is an important activity that needs to be done for needy patients or elderly people. Caregivers are very easily exposed to work-related musculoskeletal disorders. Manual patient handling, one caring task, is a major issue in healthcare sectors due to forceful exertion (handling overweight and obese patients), awkward posture (reaching, kneeling, bending, and twisting), and repetitions. The objective of this study was to analyze the body posture and the upper-limb, lower-limb, and back muscle activities when transferring patients from bed to wheelchair and wheelchair to bed manually or with the use of an assistive lift device.

The authors describe the research methodology step by step: experimental part, apparatus, experimental procedure, statistical analysis of the recorded results for MC, RC, B2C and C2B. 

I suggest to clarify the following aspects:

1) What were the selection criteria for the participants? Do the participants have the same experience in the field? The caregivers have to be men, with a certain physical condition? It is important to mention if the participants know and respect the handling procedures, if they have a good health status and in which period of a shift the study was conducted. The number of 10 subjects is small.

The authors mention that the study was approved by the Korea Institute of Industrial Technology, but it is still important to mention some regulations for this type of activity: caregivers in health-sector.

2) it is obvious that if caregivers perform their tasks with the help of RC, their workload will be significantly reduced, the prevention of occupational diseases will decrease, and their working conditions and state of health will be significantly improved. I think it is important to mention if the handling procedures are still the same or if they can be changed/improved without harming the patients. Is there a way to set new handling procedures to reduce the medical problems when the caregiver is transferring the patient from bed to wheelchair and from wheelchair to bed?

I think it is important to mention further possible research based on this article. One area of application could be the development of new ergonomic equipment that can help the caregivers from the health centres, especially those who do not have the best physical condition to perform their tasks.

Reviewer 2 Report

The study compared the effect of a robot lift assist device with manual care when helping the patient from bed to wheelchair and from wheelchair to bed. The paper is well drafted. There are some questions listed to be clarified and revised.

1. Introduction.

(1) Page 1, line 35. There is a drafting error. Two words of caregivers were written in the sentence.

(2) Page 2, line 63 and line 65. What does L5/S1 mean?

2. Methods.

(1) Page 4. 17 sensors were used for measuring the body posture angles. Where were these sensors attached on the body position? The two pictures of a male in standing posture with sensors were not clear, and there were no sensors attached on the neck of the subject. Besides, in figure 5, the caregivers were females. So, Did the subjects of the caregivers include both males and females? These were not clearly introduced in the paper.

(2) How about the patient’s condition? Were the patients fat or thin? How about their weight and height? How about their physical condition? Did they totally rely on the caregivers or they can move their legs or arms during the transfer from bed to wheelchair and from wheelchair to bed? These information should be given, since the patients’ conditions can affect the results of the study.

 3. Results. Figure 7 and Figure 10 are not clear. The numbers in the figure cannot be distinguished. What did “A” and “B” mean in these figures? Also, what did “A” and “B” mean in Figure 13?

4. Discussion. The lift assist device helped the caregivers lifting the patients from bed to wheelchair and from wheelchair to bed. Why the muscle activity in upper limbs and the back were not significantly reduced compared with those in the manual care, but the lower extremity muscle activity was significantly reduced?

Reviewer 3 Report

Abstract

-          The summary uses too many acronyms which requires reading it several times, specifically one remains unclear in the line 20: “…..ROM compared to those of MC.

Materials and Methods

-          In material and methods, I miss the 10 caregivers who participated in the study, the inclusion and exclusion criteria for them. Because there may be previous injuries that limit results or that cannot be performed correctly despite the explanations of the researcher.

Resultas

-          In each figure, the explanations previously made by the authors are similar to the results that can be seen in them. There are data that are repeated in the text and the graph and make them redundant. They could specify more briefly the most relevant result, because the rest of the percentages can be read in the graphs.

Discussion

-          In the discussion they only use two references for this, it seems to me that it would be more enriching.
